# Security Concerns in MMO Games—Analysis of a Potent Application Layer DDoS Threat

**DOI:** 10.3390/s22207791

**Published:** 2022-10-14

**Authors:** Nikola Gavrić, Živko Bojović

**Affiliations:** Faculty of Technical Sciences, University of Novi Sad, 21000 Novi Sad, Serbia

**Keywords:** distributed denial-of-service attacks, application security, graph theory, gaming, massively multiplayer online games, application-layer DDoS

## Abstract

The application layer in the Internet protocol suite offers a significant degree of freedom regarding the orchestration of distributed denial-of-service attacks due to many different and unstandardized protocols. The primary focus of defending against application-layer distributed denial-of-service attacks has traditionally been Hypertext Transfer Protocols oriented while observing individual users’ actions independently from one another. In this paper, we present and analyze a novel application-layer DDoS attack in massively multiplayer online games that utilize the cooperative efforts of the attackers to deplete the server’s or players’ bandwidth. The attack exploits in-game dependencies between players to cause a massive spike in bandwidth while the attackers’ traffic remains legitimate. We introduce a multiplayer-relations graph to model user behavior on a game server. Additionally, we demonstrate the attack’s devastating capabilities on an emulated World of Warcraft server. Lastly, we discuss flaws of the existing defense mechanisms and possible approaches for the detection of these attacks using graph theory and multiplayer-relations graphs.

## 1. Introduction

Distributed denial-of-service (DDoS) attacks have traditionally been a significant disturbance to all service providers. They meet a wide range of these attacks at different network communication layers. However, a recent prominent trend indicates the emergence of new, invisible, and increasingly sophisticated application-layer DoS attacks. These attacks aim to avoid detection while bringing the same impact as regular flooding DoS attacks [1]. DDoS attacks take advantage of firewalls and intrusion prevention system (IPS) devices to pass legitimate traffic, thus eliminating one security layer for the attacker [2]. Gaming servers are no exception to DDoS attacks [3], and some authors point to them as entry points for cybercrime activities [4]. Some reports [5] indicate that gaming servers have become the most frequent DDoS targets. Researchers and security professionals identified many DDoS attacks during the Coronavirus pandemic lockdown [6]. Gaming servers have become a vulnerable target due to many security flaws in standardized and unstandardized protocols and the different types of server applications. Moreover, gaming servers are often far more susceptible to DDoS attacks than websites because the quality of service (QoS) often requires low latency, jitter, and processing delays. Therefore, even occasional peaks in latency or processing delay can badly affect users’ quality of experience (QoE). For example, we can examine the DDOS attacks on Ubisoft’s game servers. During these attacks, the players experienced high latency or service outages, resulting in significant losses for the company [7]. World of Warcraft [8] servers have already experienced an in-game type of DDoS attack, perhaps even unintentionally, as they discovered that spatial dependencies between the players at close in-game proximity contribute to significant resource demands (Figure 1).

Figure 1 shows communications between the players (vertices) who stand physically close to each other in-game, causing them to exchange various information continuously. Suppose we display all online players as vertices in a graph representing an in-game map and quantify players’ interrelationships with weighted edges. Data links occur only between players within each other’s view range in-game (Figure 1). In that case, we assume that several players (who see one another) can form a complete subgraph and broadcast the same amount of information to each other. Such communication causes quadratic bandwidth demands in terms of the number of players, whereas players with lesser connections and isolated ones consume considerably fewer resources. Consequently, World of Warcraft implemented the Sharding system [9] (Figure 2). The Sharding system splits existing players into several disjoint sets so that players from one set cannot see players from the other sets but rather have limited interaction options. Therefore, players cannot communicate extensively with everyone around them but only with a limited set of players. Thereby, several lesser subgraphs are formed rather than a large connected one, thus reducing communication. In Figure 2, the dashed edges display connections between the players that would occur without the Sharding system, whereas the colored ones show two independent groups of players. Inspired by this phenomenon, we modeled a DDoS attack with the idea that there could be many scenarios in which intentional players’ in-game actions could cause massive spikes in bandwidth demand and that the Sharding system solves only one of those.

In this paper, we present a DDoS attack that can considerably degrade the performance of a game server or its players and cause significant losses. Because of the attack’s adjustability, the attacker can tweak the desired intensity, ranging from barely noticeable and moderate to ones that cause a service outage. Both hardware resources and a server or its clients’ bandwidth can be attack targets. Launching costs of this attack are low since it does not rely on a large botnet and expensive equipment but rather on cooperation between a small set of bots disguised as regular game players, thus making it feasible for the attacker to blackmail the server or players. We have organized this paper as follows:Section 2 presents related work in terms of attacks and relevant defense approaches.Section 3 describes the attack with its theoretical and practical aspects.Section 4 shows the DDoS attack experiment and its results.Section 5 discusses the results of our research.Section 6 consists of our conclusions and future work.

This paper studies a new application-layer DDoS attack method that, unlike volumetric attacks (e.g., Coremelt and crossfire attacks), possesses the ability to carry an amplification factor (the ratio between the data sent by the attacker and data received by the victim). For these reasons, we compare attack capabilities against current defense solutions to indicate the need for new defense mechanisms. We aim to build a dynamic system framework for modeling and developing mitigation methods. We model the attack using graph theory and introduce the concept of a multiplayer-relations graph that we use to describe players’ interrelationships within a game server. From the scientific and methodological point of view, this paper introduces the following achievements:In-game dependencies between players can result in significant bandwidth demand. Therefore, we can exploit the cooperative efforts of the attackers disguised as regular players in playing games in such a manner to maximize resource depletion while minimizing detectability.The multiplayer-relations graph can describe bandwidth-consuming relations between the players on a gaming server.The potential of this attack on an emulated game server to show the attack’s feasibility and its alignment with the attack models.We show that the current defense approaches are inefficient as they do not consider cooperation between the attackers, and we discuss possible defense solutions.

## 2. Related Work

### 2.1. Attacks

Obtaining attack vectors by using a game’s API was suggested in a review of UDP amplification attacks [10]. They exploited the computational asymmetry in a scenario where individual attackers attacked the server independently.

This attack type is a feature-based one that exploits vulnerabilities to enhance available features. Our attack manifests some of the characteristics described in the DDoS attack’s taxonomy [11], such as a low number of zombies required to perform the attack from only a single machine successfully. Stealthiness is also present because it relies on exploiting the application’s features. The taxonomy does not mention the possibility of coordinated feature-based application-layer attacks. Moreover, the authors described low-rate DDoS attacks as those that try to consume all available spots. Massively multiplayer online (MMO) games translate into logging onto too many accounts, which is impractical.

Regarding bandwidth depletion, concepts similar to ours were presented on a larger scale on the network layer in the Coremelt [12] and Crossfire [13] attacks, where cooperation between the bots resulted in sophisticated network link flooding. In the Coremelt attack, flooding occurs from the data exchange between the bots, and in the Crossfire attack, it is the data exchange between the bots and the servers behind the targeted links. Similarly, in our scenario, in the star network topology, we can observe the game server as the targeted link while its players are the final destination or bots.

The application layer offers the greatest degree of freedom compared to the other layers in orchestrating DDoS attacks. Unlike the attack mentioned above, our attack can involve more sophisticated methods, making it harder to detect.

### 2.2. Defense

When it comes to detecting malicious users on the application layer, it is necessary to discuss specific issues. Firstly, a constraint comes from MMO’s real-time nature, meaning the detection algorithm must be computationally inexpensive without affecting players’ latency. An ethical question also arises on whether it is right to sanction a player for playing the game in a manner not originally intended. Our attack is undetectable on the network layer as examining individual flows does not provide sufficient information to indicate a malicious type of cooperation between the players.

There are numerous SDN-based detection approaches to the Coremelt and Crossfire attacks. However, SDN-oriented solutions do not apply to our attack because the attackers are just as good as the regular players from the network’s aspect. Somebody’s gameplay style is not a network-related issue. In a networking sense, the server affected by this DDoS attack lacks the resources to serve its players. SDN approaches are useless as there are no actual network links under attack.

At the time being, we do not have a representative MMO game server dataset. However, we believe that signature-based detection would not help cases of coordinated attacks because a player can play the game regularly and decide to use certain features used in the attack more or less often during his play. The features used in the attacks are intended to be used, so regular players will also use them.

FineLame [14] suggests tracking resource usage on the server per request. It assumes that we can attribute the resource usage at the end of each processing phase to a single request. Then, it proceeds with K-means or other desired methods to classify requests. FineLame also assumes that we already know some expected user behavior, and it focuses only on requests that take up too many resources. In the case of our attack, resource usage per request works as intended. While the requests are taking up many more resources than initially planned, they are acceptable due to the design of the MMO games. Finally, FinaLame cannot detect coordinated attacks. In terms of practical implementation, it would have to be a very complex detection algorithm, which would have to run on the game server and track thousands of requests and player groups simultaneously.

The authors of [15] initially proposed client puzzles to defend against far simpler attacks. Nevertheless, one can expand this idea to solving puzzles whenever a suspicious activity arises. Another problem would be the GPU-offloading of such puzzle algorithms, noting that powerful GPUs are common in gaming PCs and other gaming platforms. Puzzle algorithms, GPU-offloading, and constraints were discussed in [16], making this approach very difficult. Implementing puzzles seems inapplicable in real-time applications due to an inherent delay it would cause, rendering the game unplayable and depleting one’s resources, resulting in frame-rate drops and increased power consumption, both of which are unacceptable for mobile platforms. Entropy-based metrics [17] are supposed to distinguish malicious traffic based on the packets’ information quantity.

Our attack, however, can adapt to such defense methods due to its flexibility. The authors in [18] describe methods for avoiding the entropy-based defense mechanisms that can apply to our attack. The attacker may easily play regularly for a certain period, measure all relevant statistics, and then adjust the attack parameters to fit in. Furthermore, the attacker can also adapt to the gameplay style during the attack to bypass the defense mechanisms while maximizing the attack’s share of bandwidth.

Application-layer defense models [18,19,20] are often biased towards HTTP. Such models are inapplicable because they rely on HTTP-specific functions, such as POST or GET methods, browsing behavior, or visitors’ IP address distribution. IP spoofing is unnecessary due to the authentication required to log into the game and play, meaning that source IP addresses will remain the same before and during the attack. As for applying our Attack to HTTP applications, we did not find suitable testing applications, but we do not exclude the possibility of such applications.

A defense approach presented in [21] analyzes only a static case of the Coremelt network-layer attack with a fixed amount of attackers with fixed bandwidth, which we can apply against the application-layer DDoS. Moreover, it suggests dropping some packets from the most bandwidth-intensive users. Our attack can quickly adapt to such rules by reducing the flow rate and using coordinated methods. As discussed before, these attacks’ intensity can be extremely low, blending their malicious bandwidth unnoticeably within the non-malicious one. This approach can also backfire when an individual player is targeted by many attackers, resulting in the attack’s amplification, as the server would drop the victim’s packets.

After reviewing the available literature, we could not find adequate consideration of the possible existence of the attack type that we are presenting. Consequently, there is no defense specifically designed against this attack. At the same time, it is stealthy enough to bypass current detection systems, making it a potent threat for MMO gaming servers and players.

## 3. The DDoS Attack

This section explains our attack model’s details, challenges, and requirements for launching an attack. This model assumes star topology and an MMO game server. We explain the novelty of this attack in terms of graph theory, where we introduce the term multiplayer-relations graph to simplify the model.

### 3.1. The Multiplayer-Relations Graph

Due to the complex nature of describing connections between physical equipment and logical relations between the players within the gaming server’s application that cause bandwidth usage, we introduce the multiplayer-relations graph, which facilitates the modeling of our attack. The multiplayer-relations graph is the server’s view of clients (vertices) and their interrelationships, quantified by weighted, directed edges, representing bandwidth usage. Loops in this graph describe clients’ bandwidth usage for individual needs, both upload and download, as these are API calls sent to the server and results of API calls that affect the caller. We have used loops to describe actions such as playing alone in an isolated area, obtaining responses from the server, keeping the connection alive, and sending API calls with their respective arguments (e.g., chat API and message content). A directed edge quantifies bandwidth usage (server’s upload, which is the client’s download) resulting from API calls by one vertex, affecting the targeted vertex, such as a private message from one player to another. Physically, clients are in a star topology with the server in the center. However, we must point out that the multiplayer-relations graph provides much more information. This graph is based on users’ actions affecting the server application. Due to the physical graph’s lack of information, generating a multiplayer-relations graph requires information from the server application. Assuming that we have the information about source and destination of traffic between the clients, the transition from a physical to a multiplayer-relations graph is shown in Figure 3, which illustrates insufficient insight from only observing from a networking perspective. The transform procedure from the physical graph to the multiplayer-relations graph is the following:1.Remove the vertex representing the server from the graph (Vertex 0 in Figure 3).2.All edges that were incident on vertex 0 from other vertices will become loops for their respective vertices.3.Using the information obtained from the server’s application, find the initiators (API callers) of the bandwidth represented by the remaining edge’s incident from the removed vertex 0 and connect these edges to their sources. If both the source and destination of an API call are the same, add that bandwidth to existing loops among other bandwidth where the server used to be the initiator of bandwidth (e.g., keeping the connection alive). Optionally, track how much every entity contributed to the loops in case there is a need for an inverse transform.

The inverse transform procedure is the following:1.Add a vertex representing the server to the multiplayer-relations graph.2.Replace all loops with edges incident on the server vertex from their respective vertices and add edges incident from the server’s vertex to these vertices. Using the information stored during the transform, determine the weight of these edges.3.All edges incident from the non-server vertices to other non-server vertices should have their source changed to the server vertex. This step may aggregate edges, so it is also necessary to aggregate their weights.

The transform focuses on client relations, so we removed the server to avoid loading the graph unnecessarily with the server as a proxy for client communication. Step two facilitates the observation of the graph, as we know that loops represent bandwidth arising from the clients’ overall upload in addition to necessary traffic from the server-client communication. Step three may create new edges because there can be multiple initiators of bandwidth towards a vertex. By only observing the physical graph, this information is invisible. The final step is crucial for observation, and it is what makes the multiplayer-relations graph unique. The sum of weights on the physical graph and the multiplayer-relations graph must be the same. Loops in the multiplayer-relations graph comprise both upload and download for simplicity, making it impossible to transform the multiplayer-relations graph into a physical one unless we keep track of the loops’ download and upload components. Insight into the physical graph is always available on the server and we did not find it necessary for planning DDoS attacks. Therefore, the total bandwidth at any given moment equals the sum of all weights on the graph. API calls may create or delete graph edges and affect total bandwidth demand. In terms of multiplayer-relations graphs, traditional DDoS defense approaches only consider loops, whereas other edges are neglected or do not exist in some applications (e.g., visiting a web page). However, In MMO games, such an approach is ineffective as the directed edges and their weights in the graph are highly volatile. Therefore, bandwidth usage for a given number of online players can fit a large scale of values, ranging from a graph that only consists of loops to a fully connected graph, similarly to the in-game crowd problem solved by the Sharding system [9]. The attacker aims to manipulate the multiplayer-relations graph in such a way as to maximize bandwidth usage. The number of online players is unchanged from the system’s perspective. For example, suppose that there are 30 players online and all of them broadcast with a 1 Mbps rate. In that case, there is a 30 Mbps demand for every player, meaning roughly 900 Mbps will be consumed. On the other hand, if we split the players into five groups of six players, demand will be only around 180 Mbps. Moreover, we identified a few exploits other than in-game crowding to cause such a spike in bandwidth usage, and we assume there are more such exploits.

### 3.2. Orchestrating The Attack

To launch an attack on a game server, the attacker must be familiar with the application and its protocols to estimate the impact of calling APIs that will later be used as attack vectors. We can facilitate this part if there is a way for the attacker to emulate the server and perform tests. This part is critical because a successful attack requires an appropriate set of attack vectors. Obtaining attack vectors can also be performed by directly modifying the client application, its memory, or its network packets using some widely available tools [22,23], as presented in Figure 4. If any parts of the game rely on client-sided constraints, those could be removed to amplify the attack. An example would be removing the timers for calling APIs we intend to abuse or abusing a particular protocol by disrupting its client-side components (e.g., selectively removing ACK messages). A suitable attack vector is an API call that causes substantial traffic or computational demand.

After an attack vector set has been identified, the attacker must configure bots and a system that controls and synchronizes their behavior, as illustrated in Figure 4. The final part before launching the attack involves technical details such as determining the number of bots or physical devices and used networks. Such decisions depend on the quality of the attack vectors and defense mechanisms that a game server might possess (e.g., throughput limiters per socket or IP address). The controlling application can be a separate application and an in-game feature, such as an add-on, which is a user-programmed addition to the game client. The controlling application’s purpose is to automatize and synchronize bots’ actions. For instance, many MMO games have player grouping mechanisms in which an individual player can be a group leader. Changing a group’s leader requires the leader to call an API and specify the new group leader’s name. After processing that API, the server assigns the leader role to the new player, and everybody in the group is notified about the change. In this situation, the attacker could form a group of bots that would randomly swap leader roles indefinitely. Moreover, the games usually do not include the GUI necessary to re-promote oneself. For this reason, the attacker could interfere with the network packets and change the argument of the API call to set himself as the new group leader if there is no server-side check for actions that were initially not intended to be executed.

### 3.3. Detectability

The nature of MMO games and DDoS attacks allows the application of various attacking strategies. For instance, there can be a variable number of attackers, as any bot can be disguised as a regular player for an indefinite period. Several groups of bots can work together with different attack vector sets, with each bot attacking with different intensities or switching between groups. These abilities make the detection of the attack tremendously complicated.

## 4. Experiment and Results

We emulated a World of Warcraft [8] server using the TrinityCore [24] emulator to demonstrate the attack’s feasibility. We used a list with descriptions of all WoW API calls [25] to form our attack vector. Even though there are many attacking strategies available, we stick to a relatively simple one with only one attack vector and a constant number of attackers at a constant attacking intensity for the sake of demonstration. As this is an emulated experiment, there are no players other than the attackers, so we decided to isolate them in-game as much as possible to minimize other sources of traffic in order to observed how our theoretical and experimental models compare.

### 4.1. Attack Setup

The attack vector consists of a SendAddonMessage [26] API call, which we use to send messages via add-on channels. Typically, these messages are invisible to players and uses for communication between add-ons. In the first scenario, the API call is configured to broadcast messages to all bots used in the attack to maximize bandwidth depletion. Broadcasted messages are also sent back to the API caller. We can intuitively explain this behavior by sending a message to a group chat, where the sender sees his message appearing in the chat after the server broadcasts it. In the second scenario, the bots target a single player with a simple volumetric attack that causes lesser overall bandwidth. However, it might disrupt the victim’s connection or even cause disconnection. The selected API call causes the sending of the desired command with its arguments, such as message contents and the receiving channel or player’s name to the server. After the server processes the requested API call, the add-on message is transmitted to the receiver(s) described in the API argument. The bots were controlled via an add-on that we created for the master bot to send commands via an add-on channel to the bots to execute them upon receipt. We tweaked the add-on and used an auto-clicker to precisely set the desired upload speed per bot to 0.2 MBps. A total of 20 attacking bots were involved. We present the attack schematic in Figure 5.

### 4.2. Theoretical Background

The second attack scenario is relatively simple because it is a volumetric attack, so the expected traffic that the server sends to the victim should be approximately equal to the traffic it is receiving from the attackers. In terms of the multiplayer-relations graph, the victim is a vertex with incident edges coming from the attacker’s vertices. However, a more complicated case is present in the first attack scenario, where bandwidth demands (per second) caused by the attack (denoted as *b*) by an individual attacker can be obtained as follows.
(1)$$\begin{array}{c} b=f\left(n_{i},k_{i},N\right)=\left(1+C\right)\cdot \left(S+\sum_{i=1}^{N}\left(n_{i}\cdot k_{i}\right)\right) \end{array}$$

*N* is the number of receivers of the message, $n_{i}$ is the number of messages per second, and $k_{i}$ is the size of each message being sent to the *i*-th receiver. *C* represents protocol-related messages such as other API arguments, ACK messages, and others. *S* represents uploaded data from the client to the server, which will later be transmitted to other bots (modeled as a loop as these are only API calls). However, with the chosen attack vector, we simplified the attack so that all of malicious messages have the same size. The messages are equally broadcasted at the same rate to everybody; therefore, we have the following.
(2)$$\begin{array}{c} n_{i}=n\wedge k_{i}=k\;for\;i=1,\ldots ,N \end{array}$$
(3)$$\begin{array}{c} x=n\cdot k \end{array}$$
(4)$$\begin{array}{c} S=x \end{array}$$
(5)$$\begin{array}{c} f(x,N)=(1+C)\cdot (x\cdot (N+1)) \end{array}$$

The value of *x* is now traffic at a constant rate and represents the traffic sent to a player other than the API caller. We assume that *S* and *x* are of the same size, meaning that the attacking API call (Upload to the server) with an add-on message in its arguments is equal in packet size to its counterpart, retransmitted by the server. To simplify the model further, as *C* is a relatively small number, we can ignore it; thus, we have the following.
(6)$$\begin{array}{c} C\to 0 \end{array}$$
(7)$$\begin{array}{c} f(x,N)=x\cdot (N+1) \end{array}$$

In terms of graphs, *N* is the number of edges (including a loop) involved in the attack. These edges are incident to the vertex performing the attack, and *x* is the weight of each edge. The plus one factor represents the additional weight of the attacker’s loop as the attacker’s upload data. As we perform the attack with *N* attackers, we create a complete graph with N·(N−1)-directed edges between *N* vertices, indicating quadratic bandwidth demands in terms of the number of the attackers. There are additional *N* loops for the attackers receiving their own messages from the server plus *N* loops for their API call upload. Therefore, we get an approximation of total bandwidth demand caused by attackers (denoted as *B*).
(8)$$\begin{array}{c} B=N\cdot b=N\cdot f\left(x,N\right)=x\cdot \left(N^{2}+N\right) \end{array}$$

In our attacking scenario, this means that we are supposed to roughly deplete around 80 MBps of the server’s upload and 4 MBps of its download. We can present this attack model as a complete multiplayer-relations graph with N loops. As illustrated in Figure 6 with only a few attackers, loops show the attacking messages sent to the server and ordinary player activities, whereas directed edges show messages retransmitted by the server.

As shown in Figure 7, it is difficult to determine the reasons for communication between the players by only observing the physical graph, even in the case of only a single attacker. In the case of multiple attackers, such as in Figure 6b, the physical graph would show higher numbers without any indications for their values, hence proving the multiplayer-relations graph’s usefulness.

### 4.3. Testbed

We experimented on two servers. One hosted the World of Warcraft server, while the other was running multiple client application instances. Both servers possess a dual Intel Xeon 4110 CPUs setup and 128 GB of RAM, where one that was dedicated for the players also hosts an Nvidia Titan XP GPU. The servers are connected locally with ethernet via a single 1 Gbps link.

### 4.4. Results

We used tcpdump [27] to capture all of the network’s packets on the game server. The attack lasted for only several seconds to avoid generating large log files with redundant data. We primarily focused on observing the traffic from and to the server to see how our attack model compared to the experiment. We separated the application traffic (goodput) from total traffic (throughput) to provide more accurate insights. In our graphs, we use two different time steps (10 ms and 1 s) to differentiat between precise values in time and average values over one second in order to observe the results from different perspectives. Thus, in our first experiment, we demonstrated the depletion of the server’s bandwidth.

In Figure 8, we can precisely observe that the attacking messages are causing the server to retransmit them to each of the attackers individually. This process takes time because the server requires a certain amount of bandwidth because the 1Gbps link is limited at a 1.25 MB/10 ms rate. In terms of bandwidth, the surface of the red curve should be around 20 times lesser than the surface of the blue one, which might not be intuitive, so we illustrated it in Figure 9 among the throughputs to roughly estimate the impact of other protocols.

Even though the server receives around 4 MBps of goodput, it only responds with about 75 MBps instead of the expected 80 MBps, and it happens due to averaging the measurements per second. Averaging caused certain discrepancies, such as responses being sent before the attack. Another deviation arises from the fact that the server may reply buffered messages in large packets, causing lesser communication overhead on the application layer. However, the central area proves the quadratic-like bandwidth demand that we expected to see. Our model’s numbers match the experimental results as the red curve equals x·N, and the blue one equals approximately $x\cdot N^{2}$. In a real-life scenario, the effects of such an attack could be much broader because ordinary players could unknowingly take part in it as receivers of these messages. It is not uncommon to have players grouped into much larger chats, where even a hundred players could be on the receiving end. Such a scenario would amplify the just demonstrated attack by a linear factor, notably facilitating the attacker’s efforts.

Similarly to the first attack scenario, the attackers’ combined messages cause retransmission in several lengthy packets containing many add-on messages, rather than the expected one-by-one retransmission mechanism (Figure 10). Since this is a volumetric attack, the surface below the receiving and corresponding sending traffic curves should be roughly the same. Averaged over a second (Figure 11), the graph shows that the attacker’s traffic is around 4.5 MBps.

In contrast, the victim’s traffic is around 4 MBps due to lesser communication overhead because the attackers are sending their messages one-by-one independently, while the server is retransmitting their messages in large packets. The attack also resulted in significant degradation in the victim’s quality of service, as the delay during the gameplay became more extensive, eventually rendering the game unplayable.

## 5. Discussion

In the first attack scenario, the results obtained from the experiment align with the theoretical model, hence proving the O(N^2^) bandwidth hypothesis, where N is the number of players participating in the attack. Furthermore, we demonstrated the feasibility of the attack and its devastating potential. Practically, in a real-world scenario, on a server with thousands of players online, a few dozen online players (or a single person using multiple accounts) could potentially keep the server under DDoS for an indefinite period.

In the second attack scenario, in cases where the victim’s connection is slower than the attacker’s, the attack inevitably takes 100% of the victim’s bandwidth. The attack causes a rapid degradation of QoS in-game and other applications using the Internet from the same network, ultimately leading to disconnection from the game’s server. We can treat this attack as a volumetric attack where multiple players consume a single player’s bandwidth using brute force without any amplifications. However, a notable characteristic of the attack is that the targeted player cannot know a DDoS attack is targeting him. There are no apparent indicators, so the targeted player could attribute the QoS degradation to a bad connection or hardware.

The multiplayer-relations graph has proven to be a simple and accurate tool for describing these attacks due to matching expected (modeled) and obtained bandwidths. In addition to tracking bandwidth, constructing the graph would require tracking API calls and their respective responses in a real-world environment, which could pose a difficult upgrade for some applications. Despite this, we believe that such an approach would provide important information regarding application-layer DDoS attacks. We could use this information to develop detection algorithms.

The experiment results suggest that the consequences of the attack in a real-world environment could be much worse because of the attack’s stealthiness and ability to use ordinary players as the receivers of malicious traffic, thus consuming their and, therefore, the server’s bandwidth without their knowledge. In real-life applications, these attacks would allow the selective disconnection of players or can shut down the entire server, resulting in possible cyberbullying or extortion, prompting the need to find solutions against these attacks.

## 6. Conclusions

Online gaming is omnipresent, and MMO games are a considerable part of the IT industry. The money, fame, and excitement involved in ubiquitous MMO games naturally attract hackers. Traditional application-layer DDoS attacks rely on known patterns and protocols, and there are many suggested defense approaches against such attacks. However, this research has shown that strategic cooperation between attackers who play MMO games in undesirable ways is still not considered to the required extent. For this reason, we introduced and analyzed a novel application-layer DDoS attack that exploits interrelationships between the players on the game server. Moreover, we demonstrated its effects in two scenarios on an emulated MMO game server to prove its feasibility and impact.

Additionally, we introduced the multiplayer-relations graph to display the information relevant to the attack comprehensively. We modeled the demonstrated attacks using the multiplayer-relations graph and showed that the modeled expectations accurately portrayed the attacks. Hence, we believe that this graph will contribute to the future research of application-layer DDoS attacks.

The results displayed the MMO server application’s vulnerabilities, thus indicating the need for an adequate DDoS detection system that would consider cooperative player efforts instead of solely relying on inspecting individual actions per player. The subject of our future work will be a detection system that can identify a group of players whose combined actions contribute to a significant bandwidth demand. Ideally, this system would inspect every possible group of players and find the most critical ones. However, the complexity of such a detection system is the primary issue, especially considering that there are too many ways to group players. The real-time nature of the detection systems poses an additional constraint, in addition to the hardware required for the detection algorithm. The algorithm could easily demand more hardware than the gaming server application, resulting in significant additional DDoS protection expenses, considering that this system only protects against some types of application-layer attacks. In future studies, we will attempt to create a suboptimal detection algorithm using the multiplayer-relations graph to obtain relevant information to identify most attacks originating from the displayed attack’s idea while satisfying the constraints mentioned above.

## Figures and Tables

**Figure 1 sensors-22-07791-f001:**
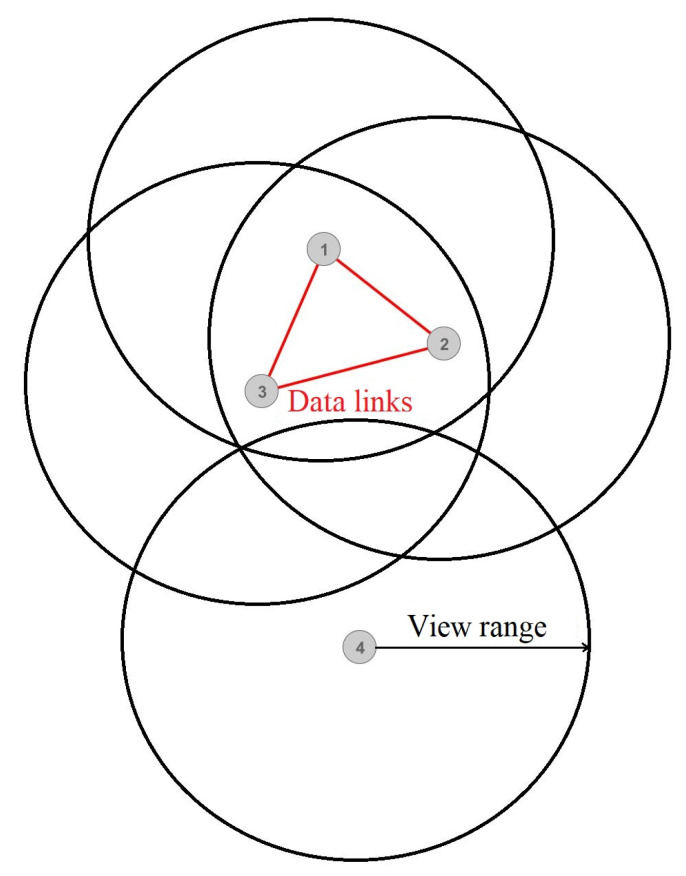
Spatial representation of the players on a game server’s map.

**Figure 2 sensors-22-07791-f002:**
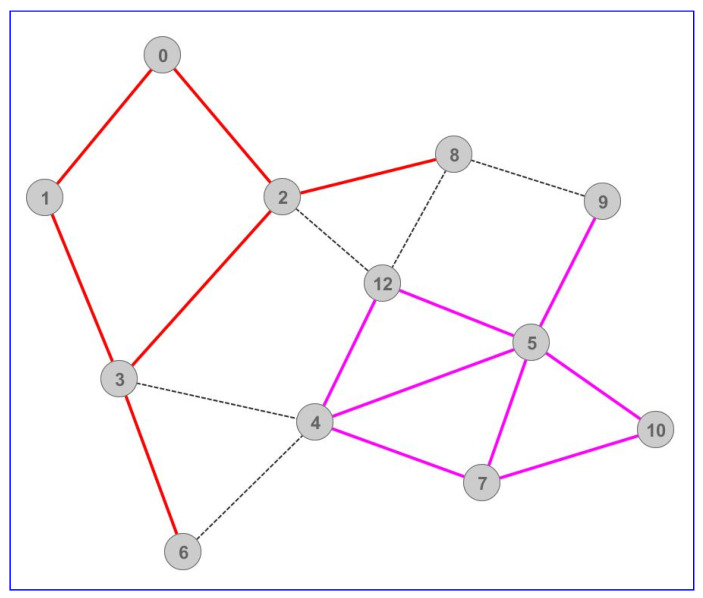
The Sharding system with two sets of players.

**Figure 3 sensors-22-07791-f003:**
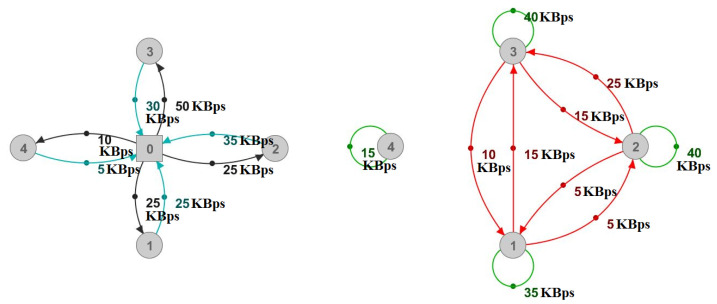
Transition from a physical graph (**left**) to a multiplayer-relations graph (**right**).

**Figure 4 sensors-22-07791-f004:**
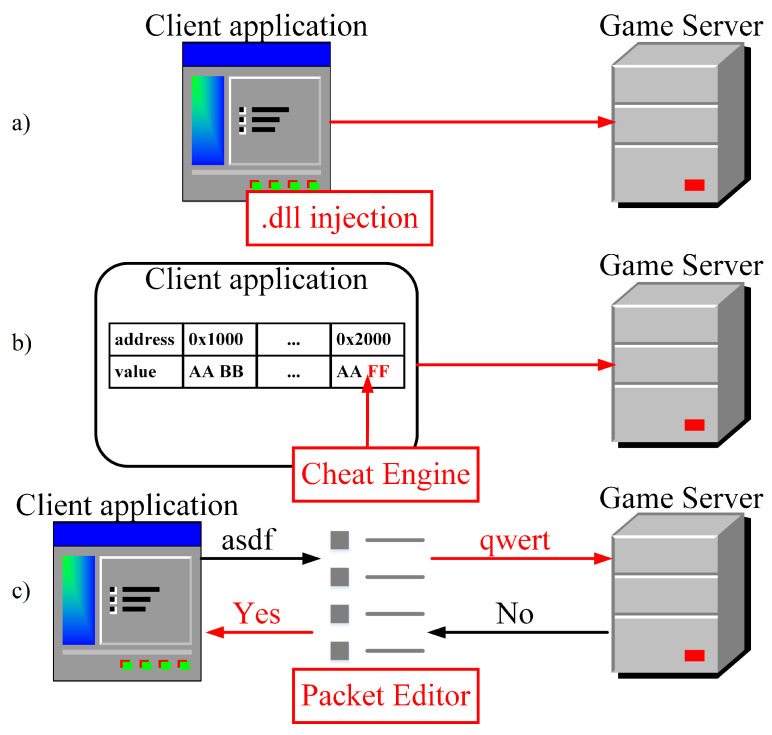
Exploitation of the client application by (**a**) modifying game files, (**b**) modifying memory during runtime, and (**c**) modifying network packets.

**Figure 5 sensors-22-07791-f005:**
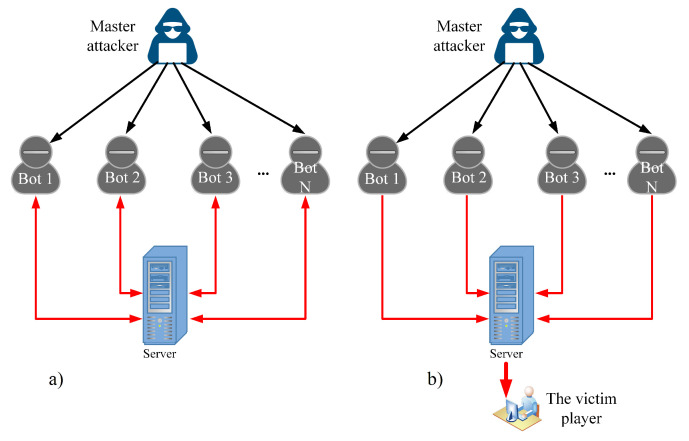
DDoS attack scheme: (**a**) attacking the server; (**b**) attacking a single player on the server.

**Figure 6 sensors-22-07791-f006:**
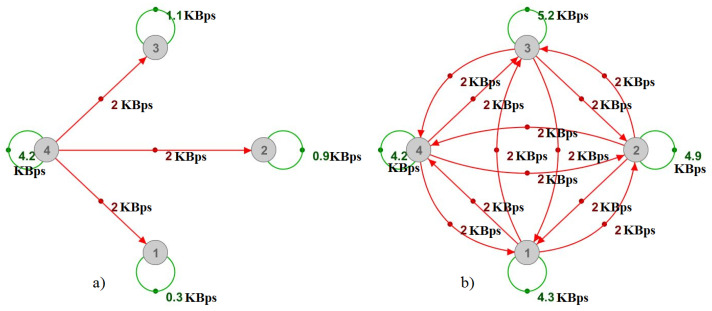
Multiplayer-relations graph—example attack model with (**a**) 1 attacker and (**b**) 4 attackers.

**Figure 7 sensors-22-07791-f007:**
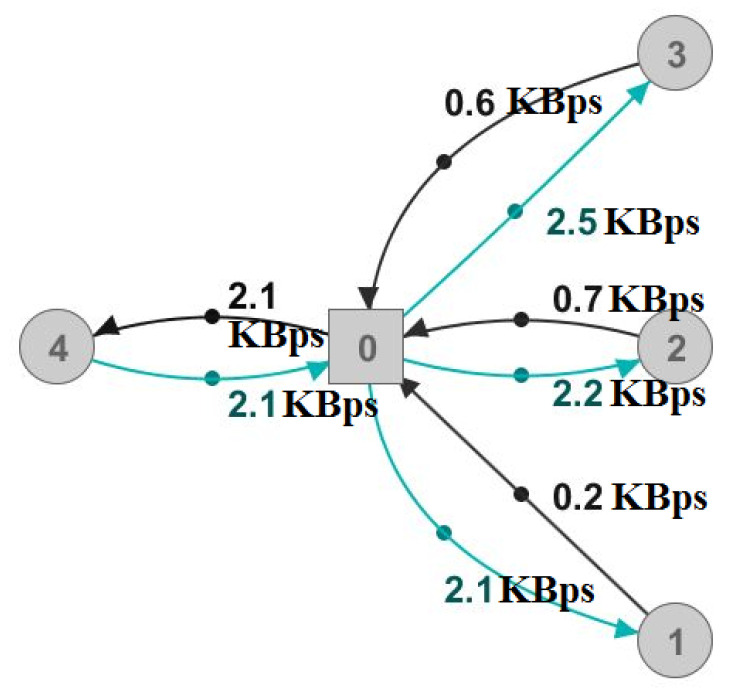
Physical representation of the multiplayer-relations graph shown in Figure 6a.

**Figure 8 sensors-22-07791-f008:**
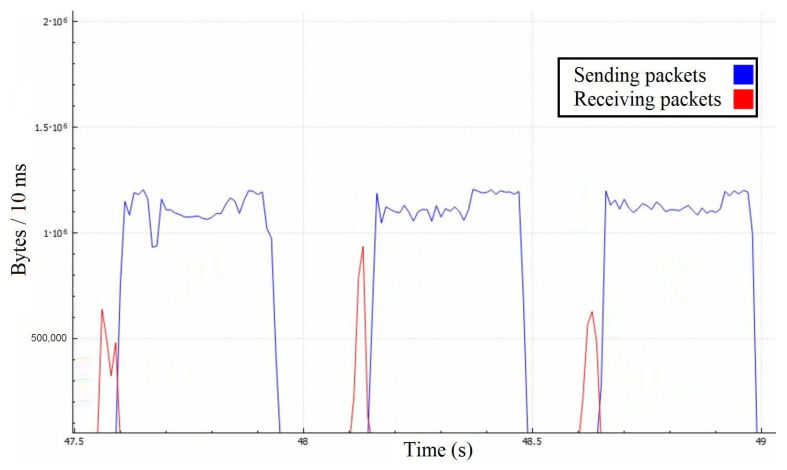
Server’s goodput in the first attack scenario with 10 ms time step.

**Figure 9 sensors-22-07791-f009:**
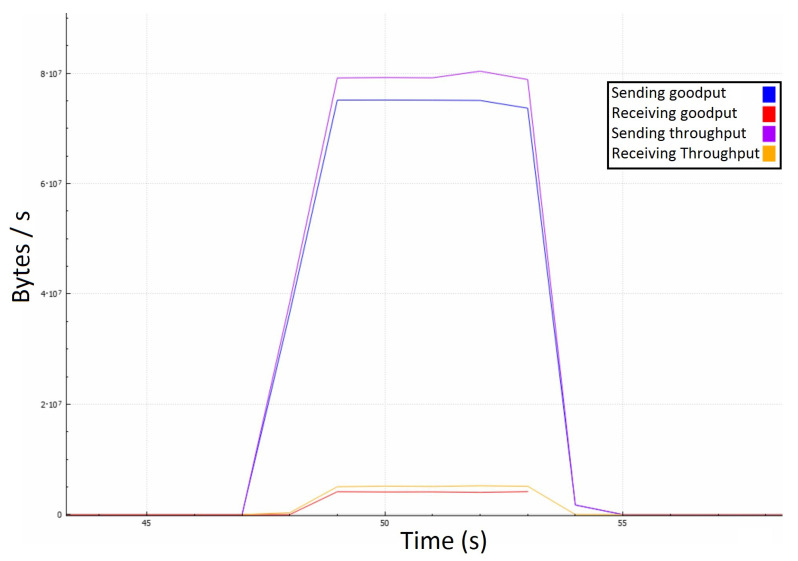
Server’s traffic in the first attack scenario with a 1 second time step.

**Figure 10 sensors-22-07791-f010:**
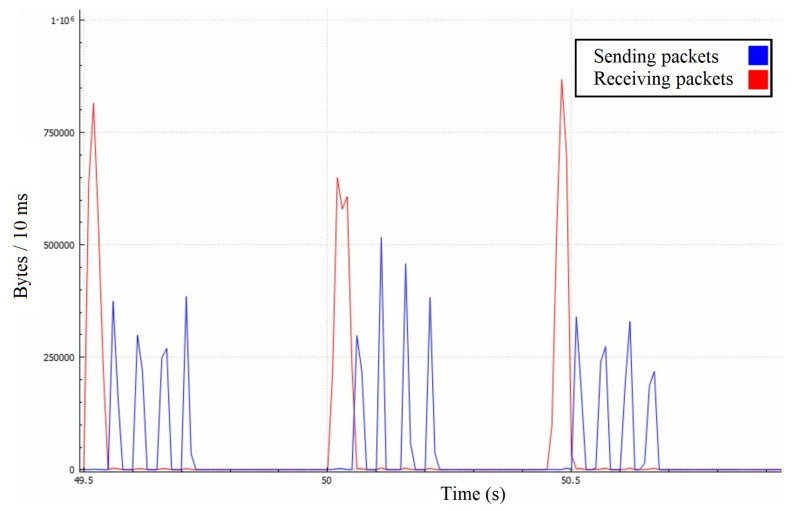
Server’s throughput in the second attack scenario with 10 ms time step.

**Figure 11 sensors-22-07791-f011:**
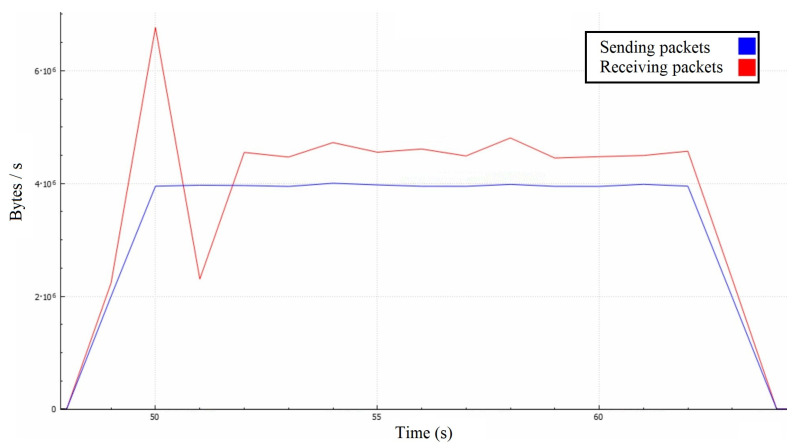
Server’s throughput in the second attack scenario with 1-second time step.

## Data Availability

Not applicable.

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
