# Peer review of "Security Concerns in MMO Games—Analysis of a Potent Application Layer DDoS Threat"

_sensors, 2022, doi:10.3390/s22207791_

Round 1
Reviewer 1 Report
I commend the research team for their manuscript.
I think the idea of using graphs to detail the source and destination of API related network flows and how they related to the experience of using the a distributed system is interesting and worth exploring further.
My initial concern for the manuscript is the creation of what the research team describes as a "logical graph":
Using Figure 3 as an example for the discussion:
1. The relationship between the edges of the physical graph and the "logical graph" requires more explanation, as does the creation of a "logical graph" from a physical graph. It is easy to assume that the unit of measurement for the edges in both graphs is total bandwidth usage. Looking at the left side of the figure for node 2 we have 35 (bandwidth usage) egress units and 25 ingress units. On the right side of the figure for node 2 we have 40 (bandwidth usage) [combined] units in a loop, but also have egress 30 units and 20 ingress units. This seems to suggest that the "logical graph" has more total bandwidth usage than the physical graph which doesn't make immediate sense.
2. I would like to know the reasoning behind the removal of node 0 from the logical graph. Whether one is looking at the TCP/IP level of the network or at the WoW level of the network, I believe node 0, as the server, is inherently part of the distributed system flow. Leaving node 0 out makes the right side of figure 3 less understandable. I believe that including node 0 would give research groups the possibilitily of utilizing multilayer graph techniques to obtain more insights for the graph you are proposing.
3. If possible, I would select a separate API to focus on in the manuscript. The API chosen is SendAddonMessage which is a grouped broadcast message type API. The referenced webpage for the API in the manuscript actually mentions that the call is server-throttled, probably because it is known that having too many recipients or many people sending to one particular client could create issues. Within proxy systems, which a WoW server inherently is, too many messages to a set of clients are generally throttled for the same reason. I believe the interesting part of your proposal, that the "logical graph" representation offers insights into application layer DDoS, is weakened due to choosing an API that is visibly problematic for DDoS attacks.
Suggestions:
The following are personal comments:
As mentioned above, I like the idea of modeling a particular API call to a graph, regardless of what measurement unit is being utilized. The name of "logical graph" leaves much to be desired, as it is not immediately clear what the graph represents. Please consider giving a more descriptive name to your proposed graph.
Possible future work:
Whatever you decide to call the graph built on the physical graph, I believe that a multilayer graph, using API as the trigger, demonstrating resource consumption other than network bandwidth would be extremely interesting. DDoS attacks can cover all resources available on a distributed system, so further work into that area would be extremely interesting to see.
Reviewer 2 Report
I have no additional comments on the work.
Author Response
I would like to kindly thank you for your review.
Reviewer 3 Report
Examination of the manuscript shows that the present investigation is not good in the present form. but, the following major points should be addressed carefully during the revision to change my decision in the revision:
- Add a nomenclature table with SI units and all used abbreviations.
- Respect the guidelines of the journal and its citation style.
- Provide a suitable reference for each used equation or model.
- The main findings should be highlighted in the abstract.
- The main objectives of the study should be itemized at the end of the introduction.
- The authors have invited to incorporate real images for the realized experiences.
- Correlate the main graphical results by an accurate relationship.
- Improve the discussion more.
- Link the title with the abstract and conclusions.
- Remove all typos and grammatical errors.
- Based on your results, how can the investigators increase the quality of these performance parameters?.
- the paper language should be revised carefully.
Reviewer 4 Report
In this article " Security Concerns in MMO Games - Analysis of a Potent Application Layer DDoS Threat " the authors have discussed possible defense mechanisms and flaws for the online attack such as exploiting in-game dependencies between 7 players. The results indicate the need for an adequate DDoS detection system that would 404 consider cooperative player efforts instead of solely relying on inspecting individual actions 405 per player. The review and results need to be improved for acceptance in this journal. The article is based on a review paper and all the suggested results should be analyzed more carefully. There is nothing new about this article in my opinion and all suggestion methods should be improved.
Round 2
Reviewer 1 Report
My concern continues to be the description of the newly named multiplayer-relations graph.
In the physical graph, it is clear that what is being measured is the network bandwidth (say the size of TCP/IP packets seen per second). I understand that the multiplayer-relations graph is connected to the physical graph, but the explanation for its creation is still not clear. As it seems that you are using the same measuring units for both graphs, it would seem to imply that if one were looking at the multiplayer-relations graph, one would expect more network bandwidth. This doesn't make sense; TCP/IP packets are either counted or they aren't. This strongly suggests that the multiplayer-relations graph is measuring something else.
I believe that the algorithm, with all its requirements and conditions, to pass from a given physical graph to a multiplayer-relations graph and vice-versa needs to be clearly defined for the paper to have merit. Please also clarify what the units are for each graph.
Reviewer 3 Report
the revised paper is okay
Author Response
Thank you for your review and beneficial suggestions.
Reviewer 4 Report
Good work
Author Response

(The authors gave the same response as above.)
